# Birds migrate longitudinally in response to the resultant Asian monsoons of the Qinghai-Tibet Plateau uplift

Wenyuan Zhang[1,2,3†‡], Zhongru Gu[1,4†], Yangkang Chen[1,5], Ran Zhang[6], Xiangjiang Zhan[1,2,7]*

[1]State Key Laboratory of Animal Biodiversity Conservation and Integrated Pest Management, Institute of Zoology, Chinese Academy of Sciences, Beijing, China; [2]Key Laboratory of Animal Ecology and Conservation Biology, Institute of Zoology, Chinese Academy of Sciences, Beijing, China; [3]Quebec Centre for Biodiversity Science, Department of Biology, McGill University, Montreal, Canada; [4]Cardiff University-Institute of Zoology Joint Laboratory for Biocomplexity Research, Chinese Academy of Sciences, Beijing, China; [5]University of the Chinese Academy of Sciences, Beijing, China; [6]Institute of Atmospheric Physics, Chinese Academy of Sciences, Beijing, China; [7]Center for Excellence in Animal Evolution and Genetics, Chinese Academy of Sciences, Beijing, China

*For correspondence:
zhanxj@ioz.ac.cn

†These authors contributed equally to this work

Present address: ‡School of Life Sciences, Nanjing University, Jiangsu, China

Competing interest: The authors declare that no competing interests exist.

## eLife Assessment

This **important** and creative study finds that the uplift of the Qinghai-Tibet Plateau – via its resultant monsoon system rather than solely its high elevation – has shifted avian migratory directions from a latitudinal to a longitudinal orientation. The authors have expanded and clarified their lines of evidence (including an enlarged tracking set and explicit caveats on species-level eBird inference), such that the central claims are now **solid**. The conclusions – that monsoon dynamics, rather than elevation per se, are most consistent with observed longitudinal reorientation – illustrate how large, community-sourced and climate-model datasets can inform continent-scale shifts in migratory behaviour over time that complement traditional approaches.

**Abstract** The uplift of the Qinghai-Tibet Plateau is one of the greatest geological events on Earth, pivotally shaping biogeographic patterns across continents, especially for migratory species that need to overcome topographical barriers to fulfil their annual circle. However, how the uplift influences animal migration strategies remains largely unclear. We compare the current flyways of 50 avian species migrating across the plateau with those reconstructed before the uplift as a counterfactual. We find that the major effect of the plateau uplift is changing avian migratory directions from the latitudinal to the longitudinal. The monsoon system generated by the uplift rather than the high elevation per se shapes those changes. These findings unveil how an important global geological event has influenced biogeographic patterns of migratory birds, yielding testable hypotheses for how observed avian distributions emerge.

## Introduction

The Qinghai-Tibet Plateau (QTP) is the most extensively elevated surface on Earth, with an average elevation of ~5 km over an area of 2.5 million km$^2$ (*Ding et al., 2022*). The uplift of the plateau

exerts profound influences on the environment and determines the biogeographic boundaries both within and across continents (*Ficetola et al., 2017*). The unique geological developments of the QTP, especially its high elevation, are believed to have influenced various taxonomic groups continuously (*Favre et al., 2015*; *Miao et al., 2022*; *Mi et al., 2021*). However, the plateau's uplift also brings up Asian monsoons, one of the most vigorous phenomena in the global climate system (*Wu et al., 2022*; *Zhang et al., 2019*). The Asian monsoons dominate large areas extending from the Indian subcontinent eastwards to Southeast and East Asia (*Yang et al., 2021*). Their evolution and variability have caused significant variations in the redistribution of water and heat via a series of natural processes, such as drought, flood, and heat waves (*Wu et al., 2022*). Given the large impacts of Asian monsoons on climate and environments, they can reconfigure the spatial patterns of biodiversity and ecosystem processes. This entails movement patterns that shape the effects of the environment on organisms (*Rubenstein and Hobson, 2004*; *Cox et al., 2022*; *Kearney et al., 2021*). However, owing to the difficulties in studying the complex effects caused by monsoons, most studies that explored the influence of the QTP just conceived the plateau as an orographic barrier (*Zhan et al., 2011*; *Zhao et al., 2023*; *Lei et al., 2014*). The role of monsoons in shaping species movement patterns remains poorly understood.

Animal movement underpins species' spatial distributions and ecosystem processes. One important animal movement behaviour is migration between breeding and wintering grounds (*Wilcove and Wikelski, 2008*; *Somveille et al., 2021*; *Zhang et al., 2023*). Those migratory journeys have motivated a body of different approaches and indicators to describe and model migration, including migratory direction, speed, timing, distance, and staging periods (*Chen et al., 2024*; *Gu et al., 2024*). Amongst them, the migratory direction is one of the most prominent indicators for migration patterns, evidenced by a majority of animals migrating latitudinally between wintering and breeding areas (*Gu et al., 2024*). This can be explained by not only the fact that wintering sites are usually located in the warmer south (e.g. Tropic) and breeding sites located in the cooler north (e.g. Arctic), but also the earth's magnetic fields that are arguably believed to affect the latitudinal migration of animals (*Guerra et al., 2014*; *Gulson-Castillo et al., 2023*; *Wynn et al., 2022*; *Takahashi et al., 2022*). However, the migratory direction can be changed from latitudinal to longitudinal when the animal faces environmental changes (*Gu et al., 2021*; *Dufour et al., 2021*; *Lehikoinen and Virkkala, 2016*; *McCaslin and Heath, 2020*; *Briedis et al., 2020*).

Environmental fluctuations in the QTP are relatively small over the *longue durée* after the final-stage uplift (*Li and Fang, 1999*), but few studies have evaluated how environmental heterogeneity across the QTP might influence the migratory behaviour of birds (but see migratory pattern descriptions, e.g. *Zhao et al., 2024*, *Pu and Guo, 2023*). Yet it remains unclear whether and how these shifts systematically alter species' migration patterns rather than a simple assumption that the QTP birds exploit resources according to their availability. Therefore, testing whether migration patterns vary consistently for birds that migrate across the QTP is key to our understanding of the processes that determine movement patterns and provides insights into how they may affect community organisation and functioning under the context of global environmental change.

In this work, we leverage community-contributed and satellite-tracking data to explore the impacts of the QTP uplift in terms of both the development of its high elevation and Asian monsoons on the migratory strategies for the birds that migrated across the plateau. We do this by reconstructing the environments before the uplift and contrasting migratory directions of 50 bird species (see *Supplementary file 1* for a full list of species) between breeding and wintering areas in environments before the uplift with those at present. Thus, the simulated environments before the uplift of the plateau serve as a counterfactual state. The use of counterfactual is important to support causation claims by comparing what happened to what would have happened in a hypothetical situation: 'If event X had not occurred, event Y would not have occurred' (*Lewis, 1973*). Recent years have seen an increasing application of the counterfactual approach to detect biodiversity change, that is, comparing diversity between the counterfactual state and real estimates to attribute the factors causing such changes, for example, *Gonzalez et al., 2023*. For example, a counterfactual is typically needed to evaluate the effectiveness of conservation interventions (*Bull et al., 2021*) and has been used to identify the role of biogeographic barriers in shaping the diversity of global vertebrates (*Williams et al., 2024*). Whilst we do not aim to provide causal inferences for avian distributional change, using the counterfactual approach, we are able to estimate the influence of the plateau uplift by detecting the changes of avian

distributions, that is, by comparing where the birds would have been distributed without the plateau to where they are currently distributed. A default assumption in the counterfactual analysis that should be carefully considered is that the species' responses to environmental change (i.e. pre- and post- the uplift of the QTP in our analysis) are conservative, and only in this way could we test the influence of changed environments on species. Here, we regard the counterfactual environments as an ideal tool to generate testable hypotheses on the role of the QTP uplift, because it allows for isolating the potential influence of the plateau's geological history on current migration routes to eliminate, to the extent possible, vagueness, as opposed to simply description of current distributions of birds.

We also calculate the migratory directions (azimuths) between adjacent stopover sites, breeding and wintering areas *en route*, and assess the relationship between migratory directions and environmental stress. Our findings yield the most comprehensive picture to date of how the QTP uplift likely shapes migratory patterns of birds, revealing insights into the challenges and opportunities for migratory birds in a changing world.

## Results and discussion

We have two major findings regarding distribution patterns and migratory directions of QTP birds. First, we developed a dynamic species distribution model (*Chen et al., 2024*) to track the weekly distribution of target species, capturing the interconnections of stopover, wintering, and breeding areas (see 'Materials and methods' for details). By contrasting their distributions before and after the

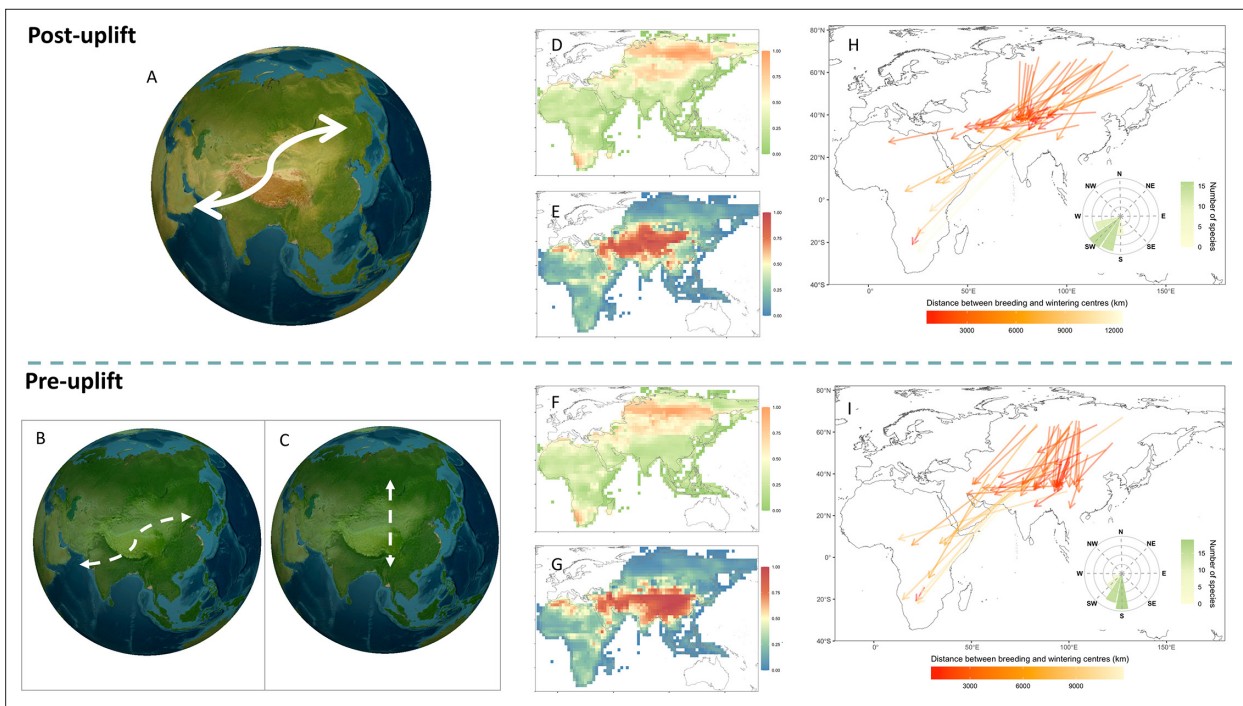

**Figure 1.** Influence of the Qinghai-Tibet uplift on avian migration strategies. (**A–C**) Schematic example of the role of Qinghai-Tibet Plateau (QTP) uplift in distribution patterns of migratory birds. (**A**) Birds migrate with a large longitudinal range in modern environments. Before the QTP uplift, birds may maintain similar migratory patterns with large longitudinal changes (**B**) or migrate with few longitudinal changes between wintering and breeding areas (**C**). The occurrence probability of 50 migratory bird species under modern environments in breeding areas (**D**) and wintering areas (**E**). The occurrence probability of birds in breeding areas (**F**) and wintering areas (**G**) before the QTP uplift. Migratory directions are identified at present (**H**) and before the uplift (**I**). The direction and length of the arrow represent migratory direction (measured by the azimuth angle) and distance from centres of breeding to wintering areas for each species. The circular barplot of the inset panel denotes the summary of migratory directions from breeding to wintering areas for each bird species, where the height and colour of the bars represent the number of species.

The online version of this article includes the following figure supplement(s) for figure 1:

**Figure supplement 1.** Birds migrate along with the vegetation gradient.

**Figure supplement 2.** Birds migrate along with the precipitation gradient.

**Figure supplement 3.** Birds migrate along with the gradient of annual temperature in the study area.

uplift, we find the distribution of migratory birds extended in longitude and narrowed in latitude with the uplift of QTP (*Figure 1E, G, and I*). Birds are more likely to migrate along a longitudinal gradient in present environments as a result of the QTP uplift (see *Supplementary file 1* for area under the curve [AUC] values for model performance of each species). Specifically, before the uplift, migratory birds had a higher probability of breeding across a vast area at low and middle latitudes on the Eurasia continent, including West Asia, Siberia, QTP regions, and even Africa, whilst their most likely breeding areas move northeastwards to the extreme north of Russia after the uplift. Different from the breeding area, the wintering area of migratory birds has a larger change in distribution probability. Birds that migrate across the QTP in the modern scenario have a higher probability of wintering in Southwest Asia and North Africa, whereas they had a higher probability of moving southeast to winter in Southern China and more areas of Africa before the uplift (*Figure 1D, F and H*).

Second, our results show that wind cost, temperature, and precipitation are three major factors that influence the overall migratory directions (both autumn and spring) of birds, despite the differences in autumn and spring migration across different geographic areas (*Figure 2*). Wind cost plays a larger role during spring migration than that during autumn migration (*Figure 2*). A higher wind cost is associated with spring migration, which suggests a greater opportunity for birds to use the wind to facilitate their flight during spring migration (*Figure 2—figure supplements 2 and 3*). They choose to follow a flyway of relatively higher annual precipitation and temperature during both spring and autumn migration (*Figure 2—figure supplements 3 and 4*). Apart from those three factors, no evidence is found for strong impacts of elevation and vegetation on the direction of migration (*Figure 2*, *Figure 2—figure supplements 1 and 6*).

Aside from the broad influences of QTP uplift, when migrating across different geographic areas, that is, areas west of (longitude < 73°E, West QTP), areas in the central (73°E ≤ longitude < 105°E, middle QTP), and areas east of the QTP (longitude ≥ 105°E, East QTP), birds diversify their preferences in environmental conditions. Despite the fact that wind cost is the most important factor for the overall spring migration, temperature is the most prominent factor in the areas east of the QTP and in the central QTP (*Figure 2*, *Figure 2—figure supplement 1*). Once they start to migrate in the regions west of the plateau (West QTP), low wind cost in the longitudinal direction and higher precipitation are priority choices for their migration (*Figure 2*, *Figure 2—figure supplement 1*). When they reach the central and east QTP, birds migrate across those areas with increasing temperatures (*Figure 1—figure supplement 3*, *Figure 2—figure supplements 1 and 3*).

Compared with spring migration, higher temperatures act as a major clue in the areas east and west of the plateau during autumn migration, whereas the vegetation outweighs other factors when birds migrate in the central plateau (*Figure 1*, *Figure 1—figure supplement 1*, *Figure 2—figure supplements 2–4*). Besides temperature, precipitation also plays a role in all stages of autumn migration. When birds migrate, they tend to follow a flyway of decreasing annual precipitation in central QTP and West QTP and an increasing one in the East QTP (*Figure 1—figure supplement 2*).

It is commonly claimed that the initiation of migration is inherently inflexible in migratory birds (*Schmaljohann and Both, 2017*), owing to the weak or insufficient responses by migratory birds to adjusting migration behaviour (e.g. migration timing and route) (*Knudsen et al., 2011*). This claim is particularly invoked for long-distance migrants, who may face greater temporal (e.g. migration timing) or physiological constraints given the varied phenologies *en route* (*Knudsen et al., 2011*). Our results show that a major change in avian migratory patterns in response to environmental change for long-distance migrants can be adjusting migration direction from the latitudinal to the longitudinal at the scale of their whole annual migration cycle. This highlights substantial changes in migratory bird distribution and their biogeographic patterns as a result of the uplift of the QTP (*Figure 1*).

One of the biggest climatic consequences of the uplift of the QTP is the development of a unique monsoon system that has shaped environments across continents (*Zhang et al., 2015*). One typical feature of Asian monsoons is seasonal climate change, comprising a dry cold winter phase and a wet warm summer phase (*Zhang et al., 2015*). Asian monsoons also consist of several sub-systems, including the northeast monsoon and the East Asian winter monsoons that dominate the weather and climate in different parts of the plateau across different geographical periods. Our results show that wind cost, temperature, and precipitation have more important impacts on avian migration than elevations in different geographic areas (*Figure 2*). This suggests that the monsoon system, rather than the high elevations of the plateau per se, is an important factor during avian migration on the plateau

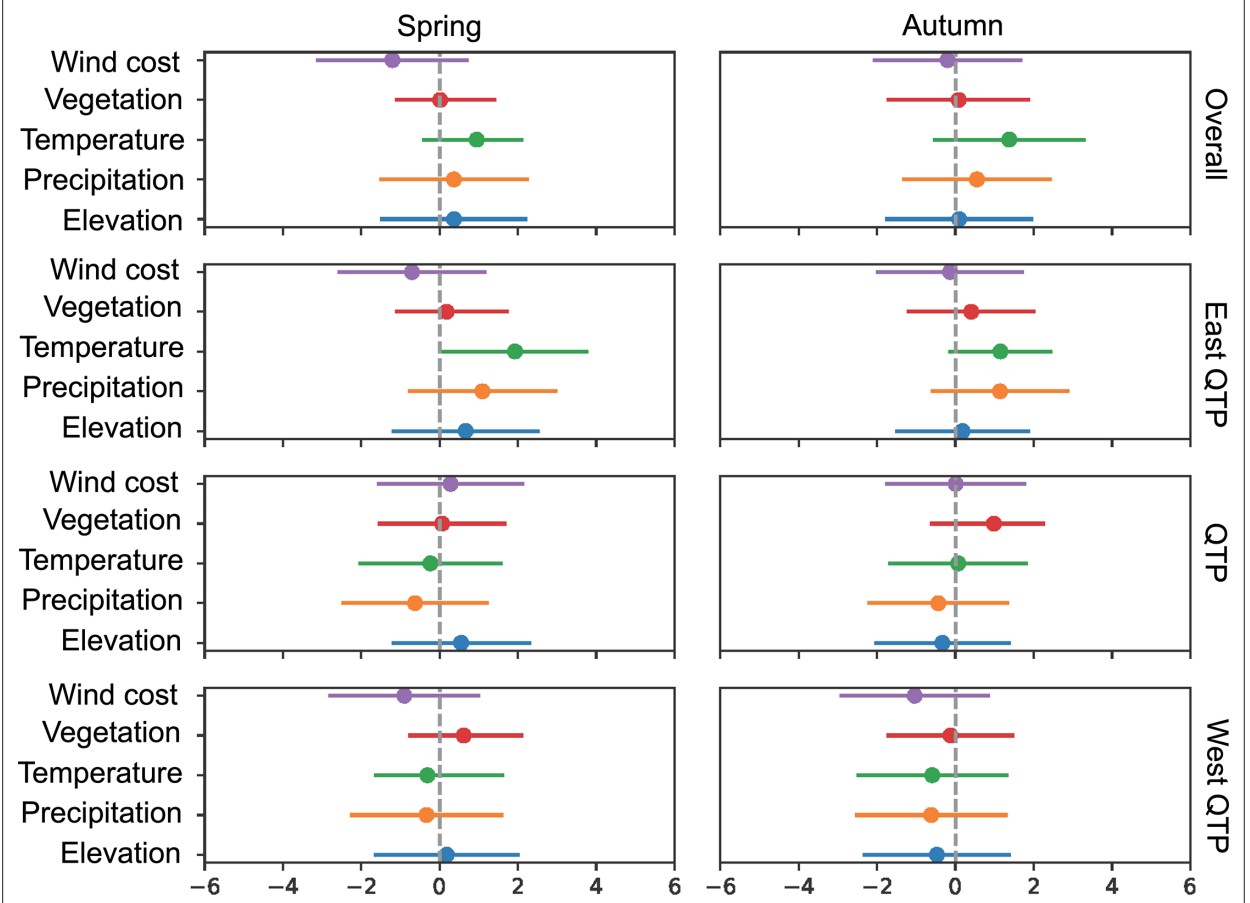

**Figure 2.** The influence of environmental factors on the direction of avian migration. Migratory directions are calculated based on the azimuths between each adjacent stopover, breeding and wintering areas for each species. We employ multivariate linear regression models under the Bayesian framework to measure the correlation between environmental factors and avian migratory directions. Wind represents the wind cost calculated by wind connectivity. Vegetation is measured by the proportion of average vegetation cover in each pixel (~1.9° in latitude by 2.5° in longitude). Temperature is the average annual temperature. Precipitation is the average yearly precipitation. All environmental layers are obtained using the Community Earth System Model. West QTP, Central QTP, and East QTP denote areas in the areas west (longitude <73°E), central (73°E ≤ longitude < 105°E), and east of (longitude ≥ 105°E) the QTP, respectively. QTP, Qinghai-Tibet Plateau.

The online version of this article includes the following figure supplement(s) for figure 2:

**Figure supplement 1.** Environmental factors that influence avian migratory strategies in the west, central, and east of the Qinghai-Tibet Plateau (QTP).

**Figure supplement 2.** Migration azimuth changes with wind cost in the area west, central, and east of the Qinghai-Tibet Plateau (QTP) during studied avian migration periods.

**Figure supplement 3.** Migration azimuth changes with average annual temperature in the area west, central, and east of the Qinghai-Tibet Plateau (QTP) during studied avian migration periods.

**Figure supplement 4.** Migration azimuth changes with average annual precipitation in the area west, central, and east of the Qinghai-Tibet Plateau (QTP) during studied avian migration periods.

**Figure supplement 5.** Migration azimuth changes with altitude in the west, central, and east of the Qinghai-Tibet Plateau (QTP) during studied avian migration periods.

**Figure supplement 6.** Migration azimuth changes with measured vegetation in the west, central, and east of the Qinghai-Tibet Plateau (QTP) during studied avian migration periods.

(*Figure 2*, *Figure 2—figure supplement 5*). Specifically, when birds begin their autumn migration in early September, the influence of the Siberian High on migration emerges as the East Asian winter monsoons start to reach the area east of the QTP, and their impacts at this stage are mainly reflected by varied temperatures and relatively less precipitation (*Gong and Ho, 2002*; *Gong et al., 2001*). This can explain why higher temperatures and increased precipitation play a more important role than wind cost in the east and central QTP (*Figure 2*) since higher annual temperatures and increased

precipitations mean more food resources for migrants (*Hoover and Schelsky, 2020*; *Jonzén et al., 2007*), whereas wind during this period is less strong than that in winter (*Gong et al., 2001*). Whilst birds migrate in the west QTP, less wind cost is becoming more important to determine their migration direction. This is, on the one hand, likely because the northeast monsoon begins to dominate the climate in the southwest of the QTP from around the end of September (*Dimri et al., 2016*). The northeast monsoon brings cold wind to sweep the QTP down towards the vast spans of the Indian Ocean (*Dimri et al., 2016*), which could facilitate the westward migration of birds. On the other hand, in the northwest of the QTP, the extended Siberia High and associated atmospheric systems that deliver cold and dry air masses to the Mediterranean surface can also provide positive wind conditions for migrants (*Labban et al., 2021*).

When migrating towards breeding grounds in spring, birds adopt strategies different from their autumn migration, accompanied by different effects of environment on their migratory directions. Temperature is more important than wind cost in spring migration (*Figure 2*, *Figure 2—figure supplement 3*). Given high temperatures usually mean relatively rich food resources for birds (*Ferger et al., 2014*; *McCain, 2009*), this suggests that birds that migrate across the QTP may focus on energy accumulation during their spring migration rather than reducing flight costs in an effort to meet the energetic demands. This pattern could be explained by a 'capital breeding' strategy — where birds rely on endogenous reserved energy gained prior to reproduction (*Stephens et al., 2009*) — rather than an 'income' strategy where birds ingest nutrients mainly collected during the period of reproductive activity (*Jönsson and Jonsson, 1997*). This aligns with studies on breeding strategies of migratory birds in Asian flyways (*Lisovski et al., 2024*). However, we note that this interpretation would require further study.

## Caveats and conclusions

Whilst we adopted both community-contributed and tracking data where potential biases existed, there are caveats to be aware of when interpreting our results. First, we adopted eBird data to infer plausible broad-scale region-to-region distributions of birds at the species level but confirming population connectivity would require targeted tracking or genetic studies. Second, we used adaptive spatiotemporal modelling to address the imbalanced distribution of sampling in eBird data, but more sampling efforts and observations are still needed in areas of sparse records to better model and predict changes of species distributions. Third, tracking data can provide detailed information of the movement patterns of species but are limited to small numbers of species due to the considerable costs and time needed. We aimed to adopt the tracking data to examine the influence of focal factors on avian migration patterns, but only 19 species, to the best of our ability, were acquired. Similar results were found in studies that used tracking data to estimate the distribution of breeding and wintering areas of birds in the plateau (e.g. *Zhao et al., 2024*; *Pu and Guo, 2023*; *Zhan et al., 2011*; *Prosser et al., 2011*; *Wang et al., 2020*; *Zhang, 2014*; *Yu et al., 2024*; *Liu et al., 2018*; *Kumar et al., 2020*). The results based on 19 species present a strong signal, but their implications could be restricted by the number of tracked species we obtained. A limited number of tracked individuals might also lead to underestimation of some migration routes. Nonetheless, an increased number of tracking techniques have been developed in recent years (including satellite tracking, computer vision, radar, and geolocator technologies), enabling us to acquire accurate information on the movements of multiple individual animals in the wild (*Nathan et al., 2022*). Future studies should build on our findings by using dedicated tracking of more individual birds and radar monitoring of animal migration over the QTP to test and investigate the influence of QTP on multiple aspects of avian migratory patterns.

Despite these caveats, our study provides a novel understanding of how QTP shapes migration patterns of birds. Albeit with the extensive influence of the plateau uplift on geology and geography, the resultant monsoon system, rather than its high elevation, is found to be a key factor shaping present avian migration patterns. Our study unveils shifts in avian migratory directions and their underlying mechanisms in the contexts of the QTP uplift, enhancing comprehension of the complex biogeographic effects on animal migration.

# Materials and methods

## Summary

We used two approaches to determine the migratory flyways of birds across the QTP. First, we quantified the distributional change of each avian species by comparing the distribution range before and after the uplift of the plateau. For the present distribution, we used a dynamic spatiotemporal abundance model – Adaptive Spatiotemporal Model (AdaSTEM) that we have developed – to obtain the seasonal distribution of birds (*Chen et al., 2024*). We then used a species distribution model (i.e. MaxEnt) to measure the correlation between present distribution and environments (*Phillips et al., 2004*). We calculated the distribution of migratory birds before the uplift of the plateau by projecting the correlation between their current distribution and environments onto environments before the uplift. Second, we obtained the specific migratory routes for each species by measuring the migratory directions (i.e. the azimuth angle between adjacent stopover sites and breeding and wintering areas) *en route*. Similarly, we used the relationships between present migratory directions and environments to predict the migratory directions pre-uplift of the plateau. Since our aim here was a prediction, we used random forest models, but we also used Bayesian multivariate regression modelling to measure the influence of environments on migratory directions of birds.

## eBird checklist

We used a community-contributed database for the dynamic spatiotemporal abundance model to measure the seasonal distribution. Specifically, we first obtained the list of bird species that might migrate across the QTP based on *Prins and Namgail, 2017*. We then requested and downloaded the eBird Basic Dataset in February 2022 (*eBird, 2022*) for 64 species. We excluded species that were not listed as 'full migrant' in BirdLife International (https://datazone.birdlife.org), which resulted in a total of 50 avian species analysed in our study and covered breeding populations in geographical Asia. We used data from the year 2019 to avoid the potential influence of the pandemic on bird observation (*Basile et al., 2021*; *Hochachka et al., 2021*) and bird behaviour (*Gordo et al., 2021*). It is worth noting that eBird provides occurrence records for species, but it generally cannot distinguish which breeding population an individual bird came from or exactly where it goes for winter. In this study, we used eBird data to infer broad-scale movement patterns (e.g. general directions and stopover regions) at the species level rather than precise one-to-one population linkages.

The eBird data may be biased by the imbalanced sampling and variation of observers' skills in identifying species. To address spatiotemporal imbalances in data distribution and the potential over-representation of birding hotspots, we conducted spatiotemporal subsampling following the method proposed (*Johnston et al., 2021*; *Fink et al., 2020*). We first assigned each checklist with a global hexagonal hierarchical geospatial indexing system (H3 system; *Brodsky, 2023*; *Jahn, 2023*), with a resolution of level 7 (~5.16 km per cell). Then, to avoid biased sampling in rare species with unusual active temporal periods, we split the 24 hours of the day into 12 equal bins and assigned a checklist to each of the bins. We then randomly subsampled only one checklist for each year–day of the year–hour bin of day–cell combination. The subsampling resulted in 5,037,088 checklists for the year 2019.

To account for the difference in observers' expertise in recognising species, we calculated the historical cumulative species count for each bird observer throughout their historical eBird checklists prior to 2019 as a proxy to measure the expertise of bird observers (*Kelling et al., 2015*). We then filtered the checklists as suggested by recent studies (*Johnston et al., 2021*; *Fink et al., 2020*):

1. Only checklists labelled as complete were included.
2. Only checklists with Traveling or Stationary protocol types were included. For checklists with the protocol type Traveling, only those with a travelling distance of less than 3 km were included.
3. The observation duration should be longer than 5 min and shorter than 300 min.
4. Observers with expertise lower than the 2.5% percentile were removed since they are less representative and may induce large bias.

## Predictor variables for spatiotemporal abundance modelling

For each remaining checklist, we extracted six types of environmental variables based on their geographical coordinates:

1. Sampling effort variables, which include protocol type, travelling distance, observation duration minutes, number of observers, and observers' expertise (measured in historical species count).
2. Temporal variables, which include day of year and observation started time of day.
3. Topographic variables, which include the mean and standard deviation of elevation, slope, north, and east aggregated within the 3 km ×3 km buffered area for each checklist. The topographic data was downloaded from EarthEnv (*Amatulli et al., 2018*) in a 1 km resolution.
4. Land cover data. We used the Copernicus Climate Change Service (C3S) Global Land Cover data with a 300 m resolution (*Climate Data Store, 2019*). We calculated the landscape variables for each of the land cover types presented in the 3 km ×3 km buffered area for each checklist, including percentage cover, patch density, largest patch index, edge density, mean patch size, standard deviation of patch size for each land cover type, and entropy across heterogeneous land cover patches.
5. Bioclimate variables. We downloaded the ERA-5 hourly data at a 0.25° resolution (*Sabater, 2019*). Hourly data of a 2 m temperature and total precipitation layer were firstly aggregated to daily level by taking the average. The day-level data were calculated using 19 bioclimate variables, which were then assigned to each checklist according to the geographical coordinates.
6. Normalised Difference Vegetation Index (NDVI). NDVI data were extracted from Terra Moderate Resolution Imaging Spectroradiometer (MODIS) Vegetation Indices 16 Day (MOD13A2) Version 6.1 product with a resolution of 16 days and 1 km (*Didan, 2021*). We further aggregated the data to hexagon level 5 based on the H3 indexing system (with edge length ~9.85 km). For each hexagon, we leveraged the pyGAM package (*Servén and Brummitt, 2018*) to apply a GAM model with 30 splines to interpolate the data to temporal ranges that were not provided by the original data. This resulted in a daily resolution dataset. We calculated six features based on NDVI and included them in subsequent modelling, that is, the median, maximum, and minimum of NDVI, and the median, maximum, and minimum of the first derivative of NDVI against day of year (sometimes referred to 'green wave') for each hexagon throughout the year.

The feature engineering resulted in 106 predictor variables, including 6 sampling effort variables, 2 temporal variables, 8 topographic variables, 19 annual climatic variables, 65 land cover variables, and 6 vegetation index-related variables. All calculations are conducted in Python version 3.9.0.

## Spatiotemporal abundance modelling

To adjust for sampling error and obtain the general migration pattern incorporating interconnections of stopover, wintering and breeding areas across species, we applied an AdaSTEM for each species to model weekly distributions of birds using stemflow package version 1.0.9.1, which we have recently reported (*Chen et al., 2024*).

AdaSTEM is a machine learning modelling framework that takes space, time, and sample size into consideration at different scales. It has been frequently used in modelling eBird data (*Fink et al., 2020*; *Fink et al., 2013*; *Johnston et al., 2015*) and has been proven to be efficient and advanced in multi-scale spatiotemporal data modelling. To briefly summarise the methodology, in the training procedure, the model recursively splits the input training data into smaller spatiotemporal grids (stixels) using the QuadTree algorithm (*Samet, 1984*). For each of the stixels, we trained a base model only using data contained by itself. Stixels were then aggregated and constituted an ensemble. In the prediction phase, stemflow queries stixels for the input data according to their spatial and temporal indexes, followed by the prediction of corresponding base models. Finally, we aggregated prediction results across ensembles to generate robust estimations (see *Fink et al., 2013* and stemflow documentation *Chen et al., 2024* for details).

We used XGBoost (*Chen and Guestrin, 2016*) as our classifier and regressor base model for its capability and balance between performance and computational efficiency. We set 10 ensemble folds, a maximum grid length threshold of 25°, a minimum grid length threshold of 5°, a temporal sliding window size of 50 DOY, and a step of 20 DOY, and required at least 50 checklists for each stixel in model training. Trained models were then used to predict on the prediction dataset with 0.1° spatial resolution and weekly temporal resolution, where the variables were annotated with the same methodology as that of the training dataset. Only spatiotemporal points with more than seven ensembles covered are predicted. In downstream analyses, we removed data points with abundance lower than 0.1 quantiles to obtain reliable predictions for each week.

## Environmental variables for species distribution modelling

Given the challenges in simulating environmental and climatic conditions before the uplift of the QTP, we modelled the environments before and after the uplift with five variables, that is, monthly wind (speed and direction), annual temperature, annual precipitation, elevation, and annual vegetation.

In detail, following *Zhang et al., 2019*, we used version 1.0.4 of the Community Earth System Model (CESM) coupled model with a dynamic atmosphere (CAM4), land (CLM4), ocean (POP2), and sea-ice (CICE4) components to simulate pre-uplift environments. CESM and its previous versions have been widely used in climate modelling, for example, *Meehl et al., 2012*, and are claimed to be capable of broadly reproducing the features of present-day climate (*Gent et al., 2011*). For CAM4, there is a horizontal resolution of ~1.9° in latitude by 2.5° in longitude and 26 layers in the vertical direction. POP2 adopts a finer grid and has a nominal 1° horizontal resolution (320 × 384 grid points, latitude by longitude) and 60 layers in the vertical direction. The land and sea-ice components share the same horizontal grids as the atmosphere and ocean components, respectively. In CLM4, multiple land surface types and plant functional types (PFTs) are contained within one grid, and CLM4 can be run in a dynamic vegetation mode to simulate natural vegetation, including trees, grass, and shrub PFTs, for example, (*Yu et al., 2014*; *Qiu and Liu, 2016*).

We initiated the modelling with two different scenarios, that is, the actual elevation and a maximum elevation of 300 m. We then used the same default preindustrial simulation for the two scenarios with a modern ice sheet, an atmospheric $CO_2$ concentration of 280 ppmv, modern orbital parameters (the year 1950), modern solar constant (1365 $W/m^2$), other atmospheric greenhouse gas concentrations ($CH_4$ and $N_2O$ set to 760 and 270 ppbv, respectively), and preindustrial aerosol conditions. We ran for 750 model years to ensure the combined atmospheric, ocean, and vegetation effects in response to the uplift of the plateau can be investigated.

## Species distribution modelling

We used maximum entropy (MaxEnt) models to compare the avian distributional change between pre- and post-uplift environments under the assumption that species tend to keep their ancestral ecological traits over time (i.e. niche conservatism). This indicates a high probability for species to distribute in similar environments wherever suitable. Particularly, considering birds are more likely to be influenced by food resources and vegetation distributions (*Martins et al., 2024*; *Qu et al., 2010*; *Li et al., 2021*), and the available food and vegetation before the uplift can provide suitable habitats for birds (*Jia et al., 2020*), we believe the findings can provide valuable insights into the influence of the plateau rise on avian migratory patterns. Having said that, we acknowledge other factors, for example, carbon dioxide concentrations (*Zhang et al., 2022*), can influence the simulations of environments and our prediction of avian distribution. MaxEnt compares the environmental features at presence points to those of pseudo absences to discriminate the suitable area (*Phillips et al., 2006*). MaxEnt builds models using a generative approach and thus has an inherent advantage over a discriminative approach, especially when the amount of training data is small (*Phillips et al., 2006*). Due to its good performance compared to other species distribution modelling techniques, MaxEnt is widely used in the study of biogeography and conservation biology.

We ran the MaxEnt model using default settings but with 1000 iterations. For each model, we ran 20 bootstrap replications, and each time 75% of locations were selected at random as training samples, while the remaining 25% were used as validation samples. We applied AUC of the receiver operator characteristic (ROC) to assess the performances of the models (*Supplementary file 1*). AUC is a threshold-independent measurement for discrimination ability between presence and random points (*Phillips et al., 2006*). When the AUC value is higher than 0.75, the model was considered to be good (*Elith et al., 2006*; *Zhang et al., 2018*).

## Migratory direction

To obtain the species list of birds that migrate across the QTP with available tracking data, we checked Movebank (movebank.org) together with literature reporting avian migratory routes across the plateau. For those who did not upload their data to Movebank, we digitalised the routes. Specifically, we built a new geographic layer with the same coordinate systems of each reported route and matched the layer with the images of routes. We then delineate migratory routes on the new

geographic layer where the geographic information of the routes was achieved. This resulted in seven representative species that migrated across the plateau.

We used the same environmental variables, except for wind, for the species distribution model to investigate the potential influence of environments on migratory directions. We calculated wind connectivity to account for the influence of wind, considering wind connectivity has been identified as a key factor driving avian flying patterns (*Kemp et al., 2010*). Since we aimed to investigate the migration patterns at large spatiotemporal scales, we measured the wind connectivity at a monthly resolution to enable analysis of seasonal differences. We adjusted the R package rWind for the computation. In detail, we replaced the default wind data from the Global Forecasting System atmospheric model with our monthly wind data from CESM as input. For both wind costs before and after the uplift of the plateau, we then calculate the movement cost from any starting cell to one of its eight neighbouring cells (Moore neighbourhood). This includes three parameters, that is, wind speed at the starting cell, wind direction at the starting cell (azimuth), and the position of the target cell. A movement connectivity map was then determined after performing the default algorithms (*Fernández-López and Schliep, 2019*). We reversed the values of cells on the connectivity map, as we aimed to investigate the influence of wind cost, whereas the map showed the importance of the cell to maintain connectivity.

We used a random forest model and a multivariate linear regression model under the Bayesian framework to analyse the influence of environments on avian migratory directions. We first used the random forest model to measure the correlation between migratory directions and modern environments and predict the migratory direction before the uplift of the plateau. We then compared the influence between modern environments and environments before the uplift using a multivariate linear regression model under the Bayesian framework. We adopted two strategies for those two modelling approaches. First, we applied regression to different combinations of season-stage separately (seasons: spring, autumn; stages: overall, east QTP, central QTP, west QTP), resulting in eight regression models. Second, we additionally included species as random variables by applying hierarchical modelling, which also resulted in eight regression models.

All variables were standardised for comparison. All Bayesian models were conducted with PyMC version 5.5 (*Salvatier et al., 2016*) in Python version 3.9.0 environment. We used a NUTS sampler with a numpyro backend (jax.sample_numpyro_nuts) to run four chains, each with 3000 tuning and 3000 posterior chain sampling. We assessed the model convergence using potential scale reduction factor (Rhat) and effective sample size (ESS), where all parameters in all models met the criteria of Rhat <1.03 and ESS >400.

## Acknowledgements

This work was supported by the CAS Project for Young Scientists in Basic Research (YSBR-097) and the National Natural Science Foundation of China (nos. T2450075, 32125005, 32270455,31821001).

## Additional information

### Funding

| Funder | Grant reference number | Author |
| --- | --- | --- |
| Chinese Academy of Sciences | YSBR-097 | Zhongru Gu<br>Xiangjiang Zhan |
| National Natural Science Foundation of China | 32270455 | Zhongru Gu |
| National Natural Science Foundation of China | 32125005 | Xiangjiang Zhan |
| National Natural Science Foundation of China | 31821001 | Xiangjiang Zhan |
| National Natural Science Foundation of China | T2450075 | Zhongru Gu<br>Xiangjiang Zhan |

| Funder | Grant reference number | Author |
| --- | --- | --- |

The funders had no role in study design, data collection and interpretation, or the decision to submit the work for publication.

## Author contributions

Wenyuan Zhang, Conceptualization, Data curation, Formal analysis, Validation, Visualization, Methodology, Writing – original draft, Writing – review and editing; Zhongru Gu, Data curation, Formal analysis, Validation, Visualization, Methodology, Writing – review and editing; Yangkang Chen, Data curation, Formal analysis, Visualization; Ran Zhang, Data curation, Formal analysis; Xiangjiang Zhan, Conceptualization, Resources, Supervision, Validation, Investigation, Methodology, Writing – original draft, Project administration, Writing – review and editing, Funding acquisition

## Author ORCIDs

Wenyuan Zhang (iD) https://orcid.org/0000-0002-7102-9922
Zhongru Gu (iD) http://orcid.org/0000-0002-0178-1107
Xiangjiang Zhan (iD) https://orcid.org/0000-0002-4517-1626

Joint Public Review: https://doi.org/10.7554/eLife.103971.4.sa1
Author response https://doi.org/10.7554/eLife.103971.4.sa2

# Additional files

## Supplementary files

Supplementary file 1. List of species analysed in our study and AUC values for modelling their distributions.

MDAR checklist

## Data availability

All data, code, and materials used in the analysis are available from GitHub (https://github.com/plmyann/QTPbirds copy archived at *plmyann, 2025*).

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
