## [Editor Report · eLife Assessment]

This **important** and creative study finds that the uplift of the Qinghai-Tibet Plateau – via its resultant monsoon system rather than solely its high elevation – has shifted avian migratory directions from a latitudinal to a longitudinal orientation. The authors have expanded and clarified their lines of evidence (including an enlarged tracking set and explicit caveats on species-level eBird inference), such that the central claims are now **solid**. The conclusions – that monsoon dynamics, rather than elevation per se, are most consistent with observed longitudinal reorientation – illustrate how large, community-sourced and climate-model datasets can inform continent-scale shifts in migratory behaviour over time that complement traditional approaches.

---

## [Referee Report · Joint Public Review]

The study assesses how the rise of the Qinghai-Tibet Plateau affected patterns of bird migration between their breeding and wintering sites.

This is an interesting topic and a novel theme. The visualisations and presentation are to a very high standard. The Introduction is very well-written and introduces the main concepts well, with a clear logical structure and good use of the literature. The Methods are detailed and well-described, and written in such a fashion that they are transparent and repeatable.

Editorial note: These latest revisions are minor in the sense that they expand on the dataset but do not change the primary results.

---

## [Author Response]

The following is the authors’ response to the previous reviews

**Reviewer #1 (Public review):**
The authors have done a good job of responding to the reviewer's comments, and the paper is now much improved.

Again, we thank the reviewer for positive comments during review.

**Reviewer #2 (Public review):**
I would like to thank the authors for the revision and the input they invested in this study.

We are grateful for your thoughtful feedback and enthusiasms, which helps us improve our manuscript.

With the revised text of the study, my earlier criticism holds, and your arguments about the counterfactual approach are irrelevant to that. The recent rise of the counterfactual approach might likely mirror the fact that there are too many scientists behind their computers, and few go into the field to collect in situ data. Studies like the one presented here are a good intellectual exercise but the real impact is questionable.

We understand your concern about the relevance of the counterfactual approach used in our study. Our intent in using a counterfactual scenario (reconstructing migration patterns assuming pre-uplift conditions on the QTP) was to isolate the potential influence of the plateau’s geological history on current migration routes. Similar approach was widely used to estimate how biogeographic barriers facilitated the divergent vertebrate communities across the world (e.g., Williams et al. 2024). We agree that such an approach must be used carefully. In the revision, we have explicitly clarified why this counterfactual comparison is useful – namely it provides a theoretical baseline to test how much the QTP’s uplift (and the associated monsoon system) might have redirected migration paths (Gilbert and Lambert 2010, Sanmartín 2012, Bull et al. 2021). We acknowledge that the counterfactual results are theoretical and have explicitly emphasised the assumptions involved (i.e., species–environment relationships hold between pre- and post- lift environments) in the main text (Lines 91- 98). Nonetheless, we defend the approach as a valuable study design: it helps generate testable hypotheses about migration (for instance, that the plateau’s monsoon-driven climate, rather than just its elevation, introduces an east–west shift en route).

References:

Bull, J. W., N. Strange, R. J. Smith, and A. Gordon. 2021. Reconciling multiple counterfactuals when evaluating biodiversity conservation impact in social-ecological systems. Conservation Biology 35:510-521.

Gilbert, D., and D. Lambert. 2010. Counterfactual geographies: worlds that might have been. Journal of Historical Geography 36:245-252.

Sanmartín, I. 2012. Historical Biogeography: Evolution in Time and Space. Evolution: Education and Outreach 5:555-568.

Williams, P. J., E. F. Zipkin, and J. F. Brodie. 2024. Deep biogeographic barriers explain divergent global vertebrate communities. Nature Communications 15:2457.

All your main conclusions are inferred from published studies on 7! bird species. In addition, spatial sampling in those seven species was not ideal in relation to your target questions. Thus, no matter how fancy your findings look, the basic fact remains that your input data were for 7 bird species only! Your conclusion, “our study provides a novel understanding of how QTP shapes migration patterns of birds” is simply overstretching.

We appreciate the reviewer’s comment here. We would like to clarify that our conclusions regarding longitudinal shifts in migratory distributions are based on distribution models derived from eBird data of 50 species, not merely on migration tracks from seven species. These species-level spatiotemporal models allow us to infer large-scale biogeographic patterns across the Qinghai-Tibet Plateau (QTP).

The original seven tracking species were used specifically for analysing the relationship between migration directions (azimuths) and environmental variables, offering independent support for the patterns revealed in the eBird-based distribution models. Recognising the reviewer’s concern on sample size and coverage, we have now expanded this part by incorporating migration tracks from 12 additional species, derived through georeferenced digitisation of published migratory maps. Importantly, this expansion did not change our conclusions, i.e., the monsoons instead of the high elevations act as a prominent role in shaping the current migration direction of birds in the QTP. While the overall conclusion remains unchanged, the expanded dataset led to slight changes in difference between spring and autumn migration. We have updated the Figure 2 and the corresponding results and conclusions throughout the manuscript. We have also clarified in the Discussion that regions of the QTP with relatively less data might lead to underestimation of some migration routes to make sure readers are aware of these data limitations (Lines 211-218).

The way you respond to my criticism on L 81-93 is something different than what you admit in the rebuttal letter. The text of the ms is silent about the drawbacks and instead highlights your perspective. I understand you; you are trying to sell the story in a nice wrapper. In the rebuttal you state: “we assume species' responses to environments are conservative and their evolution should not discount our findings.” But I do not see that clearly stated in the main text.

Thanks, as suggested we have clearly stated the assumptions of niche conservatism in the Introduction (Lines 91-98).

In your rebuttal, you respond to my criticism of "No matter how good the data eBird provides is, you do not know population-specific connections between wintering and breeding sites" when you responded: ... "we can track the movement of species every week, and capture the breeding and wintering areas for specific populations" I am having a feeling that you either play with words with me or do not understand that from eBird data nobody will be ever able to estimate population-specific teleconnections between breeding and wintering areas. It is simply impossible as you do not track individuals. eBird gives you a global picture per species but not for particular populations. You cannot resolve this critical drawback of your study.

We agree that inferring population-specific migratory connections (teleconnections) from eBird data is challenging and inherently limited. eBird provides occurrence records for species, but it generally cannot distinguish which breeding population an individual bird came from or exactly where it goes for winter. Our objective is not to determine one-to-one migratory links between specific populations, but to identify general broad-scale directional shifts when birds cross the QTP during their migration. We regret any confusion caused by our earlier wording. To make this clearer, we have now emphasised that our interests focus on the migratory direction and their environmental correlates, rather than population assignments. We have also rephrased the relevant text to explicitly clarify that our study operates at the species level and at large spatial scales (Lines 253–257). We exemplify how distribution of eBird observations and GPS tracking data of four species can be different from each other whilst showing similar migration patterns (Figure S10). We have also explicitly stated in the Discussion that confirming population connectivity would require targeted tracking or genetic studies, and that our eBird-based analysis could only suggest plausible routes and region-to-region linkages (Lines 200-202).

I am sorry that you invested so much energy into this study, but I see it as a very limited contribution to understanding the role of a major barrier in shaping migration.

We thank the reviewer’s honest assessment and understand the concern regarding the scope of our contribution. Our intention was not to provide an exhaustive account of all aspects of the QTP as a migratory barrier, but to address a specific and underexplored question: how the uplift of the plateau and the resulting monsoon system may have influenced the orientation of avian migration routes. By integrating both satellite tracking and community-contributed data, we have explored how the uplift of the QTP could shape avian migration across the area. We believe our findings provide important insights of how birds balance their responses to large-scale climate change and geological barrier, which yields the most comprehensive picture to date of how the QTP uplift have shaped migratory patterns of birds. We have also discussed the study’s limitations – including the small number of tracking species (Lines 205218), the use of occurrence data as a proxy for breeding and wintering regions (Lines 200-202), the uneven sampling coverage in the QTP (Lines 202-205) and the assumptions behind the counterfactual scenario (Lines 91-98). This ensures that readers understand the context and constraints of our findings.

My modest suggestion for you is: go into the field. Ideally use bird radars along the plateau to document whether the birds shift the directions when facing the barrier.

We thank the reviewer for this suggestion. We agree that radar holds promise for understanding certain aspects of bird migration, particularly for detecting flight intensity, altitudes, and timing. However, the radar systems are currently challenging to resolve migration at the level of species, populations, or individuals, which are central to questions of migratory connectivity and route selection. Most radar signals cannot distinguish between species in mixed flocks, nor can they link breeding and wintering sites for tracked individuals. In addition, the spatial coverage of radar installations remains limited, especially across remote and high-elevation regions like the Qinghai-Tibet Plateau, where infrastructure and continuous power supply are still logistically prohibitive.

The eBird dataset used in our study is itself a form of field-based observation, contributed by tens of thousands of birdwatchers across continents, including the QTP region (Figure S11). While eBird cannot provide individual-level tracking, it captures spatiotemporal patterns of occurrence at broad scales, making it a valuable complement to satellite tracking data. We would also emphasis that our team has extensive field experience in the Qinghai-Tibet Plateau (about twenty years), including multi-year expeditions to deploy satellite tags and observe migration at stopover sites.

We agree that more direct tracking (e.g. GPS tagging) would be an ideal way to validate migration pathways and population connectivity. Using the satellite-tracking data, we have showed that most tracking species shifted their migration direction when facing the QTP (Figure S6). In this revision, as stated we managed to add a number of 12 more species with satellite tracking routes. We have also noted that future studies should build on our findings by using dedicated tracking of more individual birds and monitoring of migration over the QTP. We have cited recent advances in these techniques and suggested that incorporating more tracking data could further test the hypotheses generated by our work (Lines 205-218).

**Reviewer #2 (Recommendations for the authors):**
L55 "an important animal movement behaviour is.." Is there any unimportant animal movement? I mean this sentence is floppy, empty.

We used this sentence to introduce migration. We have removed “important” to reduce ambiguous phrasing.

L 152-154 This sentence is full of nonsense or you misinterpretation. First of all, the issue of inflexible initiation of migration was related to long-distance migrants only! The way you present it mixes apples and oranges (long- and short-distance migrants). It is not "owing to insufficient responses" but due to inherited patterns of when to take off, photoperiod and local conditions.

We stated that this claim is invoked for long-distance migrants before this sentence and have rewritten the sentence to highlight that this interpretation is for long-distance migrants.

L 158 what is a migration circle? I do not know such a term.

We have amended it as “annual migration cycle”, which is a more common way to describe the yearly round-trip journey between breeding and wintering grounds of birds.

L 193 The way you present and mix capital and income breeding theory with your simulation study is quite tricky and super speculative.

We thank the reviewer for raising this important concern. We have presented this idea as an inference rather than a conclusion: “This pattern could be consistent with a ‘capital breeding’ strategy — where birds rely on endogenous reserved energy gained prior to reproduction — rather than an ‘income’ strategy where birds ingest nutrients mainly collected during the period of reproductive activity. This collaborates with studies on breeding strategies of migratory birds in Asian flyways. However, we note that this interpretation would require further study.” By adding this caution, we made it clear that we are not asserting this link as proven fact, only suggesting it as one possible explanation. We have also doublechecked that the rest of the discussion around this point is framed appropriately. Moreover, to help illustrate why we raised this ecological interpretation, we would also draw attention to examples of satellite tracking points from several species (e.g., Beijing Swift, Demoiselle Crane) in the following, which show obvious shifts in migratory direction near the QTP region. These turning points suggest potential behavioral responses to environmental constraints, such as climatic corridors or energy availability, which could help motivate our discussion of possible capital breeding strategies in these species.